# Dimeric Product of Peroxy Radical Self-Reaction Probed with VUV Photoionization Mass Spectrometry and Theoretical Calculations: The Case of C_2_H_5_OOC_2_H_5_

**DOI:** 10.3390/ijms24043731

**Published:** 2023-02-13

**Authors:** Hao Yue, Cuihong Zhang, Xiaoxiao Lin, Zuoying Wen, Weijun Zhang, Sabah Mostafa, Pei-Ling Luo, Zihao Zhang, Patrick Hemberger, Christa Fittschen, Xiaofeng Tang

**Affiliations:** 1Anhui Institute of Optics and Fine Mechanics, Hefei Institutes of Physical Science, Chinese Academy of Sciences, Hefei 230031, China; 2Science Island Branch, Graduate School, University of Science and Technology of China, Hefei 230026, China; 3Univ. Lille, CNRS, UMR 8522-PC2A–Physicochimie des Processus de Combustion et de I’Atmosphère, F-59000 Lille, France; 4Institute of Atomic and Molecular Sciences, Academia Sinica, Taipei 10617, Taiwan; 5Laboratory for Synchrotron Radiation and Femtochemistry, Paul Scherrer Institute, CH-5232 Villigen, Switzerland

**Keywords:** dimeric product, ethyl peroxy radical, self-reaction, photoionization mass spectrometry, theoretical calculation, photoelectron photoion coincidence

## Abstract

Organic peroxy radicals (RO_2_) as key intermediates in tropospheric chemistry exert a controlling influence on the cycling of atmospheric reactive radicals and the production of secondary pollutants, such as ozone and secondary organic aerosols (SOA). Herein, we present a comprehensive study of the self-reaction of ethyl peroxy radicals (C_2_H_5_O_2_) by using advanced vacuum ultraviolet (VUV) photoionization mass spectrometry in combination with theoretical calculations. A VUV discharge lamp in Hefei and synchrotron radiation at the Swiss Light Source (SLS) are employed as the photoionization light sources, combined with a microwave discharge fast flow reactor in Hefei and a laser photolysis reactor at the SLS. The dimeric product, C_2_H_5_OOC_2_H_5_, as well as other products, CH_3_CHO, C_2_H_5_OH and C_2_H_5_O, formed from the self-reaction of C_2_H_5_O_2_ are clearly observed in the photoionization mass spectra. Two kinds of kinetic experiments have been performed in Hefei by either changing the reaction time or the initial concentration of C_2_H_5_O_2_ radicals to confirm the origins of the products and to validate the reaction mechanisms. Based on the fitting of the kinetic data with the theoretically calculated results and the peak area ratios in the photoionization mass spectra, a branching ratio of 10 ± 5% for the pathway leading to the dimeric product C_2_H_5_OOC_2_H_5_ is measured. In addition, the adiabatic ionization energy (AIE) of C_2_H_5_OOC_2_H_5_ is determined at 8.75 ± 0.05 eV in the photoionization spectrum with the aid of Franck-Condon calculations and its structure is revealed here for the first time. The potential energy surface of the C_2_H_5_O_2_ self-reaction has also been theoretically calculated with a high-level of theory to understand the reaction processes in detail. This study provides a new insight into the direct measurement of the elusive dimeric product ROOR and demonstrates its non-negligible branching ratio in the self-reaction of small RO_2_ radicals.

## 1. Introduction

Organic peroxy radicals (RO_2_) are key intermediates in tropospheric chemistry and play central roles in atmospheric oxidation reactions [1,2]. They are mainly formed in the atmosphere from the addition reaction of oxygen with alkyl radicals (R), initiated by a set of oxidants, such as OH, O_3_, Cl and NO_3_. Their subsequent chemistry is very rich and exerts a controlling influence on the cycling of atmospheric reactive radicals and the production of secondary pollutants, such as ozone and secondary organic aerosols (SOA) [1,2,3]. Concretely, under the high NOx (NO and NO_2_) conditions, the reaction of RO_2_ with NO usually dominates among the various reactions of RO_2_ and contributes to regional air pollution problems through the following photolysis of the product NO_2_ to generate the ozone. In the clean atmosphere (NOx < ~20 pptv), the self-reactions of RO_2_ radicals and their cross reactions with HO_2_, other RO_2_ or OH radicals are the main loss processes.

It is well-established that the self-reactions of RO_2_ radicals mainly generate either the corresponding alkoxy radicals (RO) or an alcohol product (ROH) in combination with a carbonyl compound (R_-H_=O) [1,2].
RO_2_ + RO_2_ → RO + RO + O_2_(1)
         → ROH + R_-H_=O + O_2_(2)

Recently, isomerization reactions of RO_2_ radicals, i.e., via an intramolecular H-atom transfer or an endo-cyclization of unsaturated RO_2_ radicals, have been determined and result in higher-functionalized RO_2_ radicals after successive O_2_ addition [4,5]. In particular, with the rapid development of mass spectrometric techniques, highly oxygenated organic molecules (HOM) have been observed in experiments and many of them are formed from the self-reactions of higher-functionalized RO_2_ radicals, according to the dimeric product pathway (3) [6,7,8].
RO_2_ + RO_2_ → ROOR + O_2_(3)

The reaction rates of the dimeric product formation are measured to be high for RO_2_ radicals bearing functional groups, even up to 10^−10^ cm^3^ molecule^−1^ s^−1^, and can compete with those of their corresponding reactions with NO and HO_2_ [4]. Due to their lower vapor pressure, the dimeric products, ROOR, formed from the self-reaction of RO_2_ radicals, are believed to be essential precursors of new particles and SOA [3,6]. Consequently, understanding the dimeric product formation is crucial to explaining the missing sources of SOA and has been the subject of numerous experimental and theoretical studies [3,9].

In comparison to higher-functionalized RO_2_ radicals, the dimeric product from the self-reactions of small or primary RO_2_ radicals is considered to be negligible [10]. However, although with a small branching ratio, usually at several percent, the overall production of the dimeric product from the self-reactions of small RO_2_ radicals is not negligible considering their high concentrations in the atmosphere. For example, methyl peroxy radicals, CH_3_O_2_, have the highest concentration and can account for almost half of the total amount of peroxy radicals in the atmosphere [11]. Other small RO_2_ radicals, such as C_2_H_5_O_2_ and C_3_H_7_O_2_, etc., should also have high concentrations [12,13].

The self-reactions of small RO_2_ radicals to produce the dimeric product ROOR have attracted attention in recent years. For example, Zhang et al. carried out a theoretical study on the mechanism and kinetics for the self-reaction of C_2_H_5_O_2_ radicals and predicted that C_2_H_5_O, CH_3_CHO and C_2_H_5_OH are the major products, whereas the dimeric product C_2_H_5_OOC_2_H_5_ should be a minor pathway [14]. To explain the production routes of ROOR, Lee et al. theoretically predicted that the self-reaction of RO_2_ first proceeds through a singlet RO_4_R tetroxide structure, which undergoes two separate bond-cleavage reactions to form a singlet ^1^(RO···O_2_···RO) “cage”, where the weakly bound O_2_ is in its triplet ground state and the two alkoxy radicals have parallel spins [15]. Valiev et al. performed theoretical calculations and explored one potentially competitive pathway involving the initial formation of the triplet alkoxy radical pairs, ^3^(RO···RO), followed by rapid intersystem crossings (ISC) to the singlet surface, permitting subsequent recombination to ROOR [16]. Hasan et al. systematically investigated the above three self-reaction channels of small RO_2_ radicals with high level of theory and predicted non-negligible branching ratios for all three channels [17].

In experiments, due to the limitation of analytical techniques to directly measure the dimeric products [18], only a few results can be found in the literature and even some of them are in controversy. For instance, for the self-reaction of CH_3_O_2_, the branching ratio of the dimeric product CH_3_OOCH_3_ channel was measured at 7% by Weaver et al. [19] and at 8% by Niki et al. [20]. The branching ratio of C_2_H_5_OOC_2_H_5_ in the C_2_H_5_O_2_ self-reaction was measured by several groups, with its value estimated at 6% or 9%, with some studies finding no evidence for the peroxide formation [21,22,23]. The current IUPAC database therefore recommends, for the C_2_H_5_O_2_ self-reaction, based on the analysis of stable end products, a yield of 0.63 for the alkoxy path (1), 0.37 for the molecular path (2) and 0 for the dimer path (3) [24]. More recent studies directly measuring the radical yield (1) from the self-reaction of C_2_H_5_O_2_ found only a minor yield of 0.28 [25] and 0.31 [26] for the alkoxy path (1), in strong disagreement with the IUPAC recommendation. A possible explanation for this strong disagreement could be a higher yield of the dimer path (3), which could, due to its instability, appear in the stable end-product studies as a radical path. In addition, the structures of the dimeric products still remain elusive [8].

This motivated us to reinvestigate the self-reaction of C_2_H_5_O_2_ radicals by using a vacuum ultraviolet (VUV) lamp-based photoionization mass spectrometer in Hefei [27,28] and a double imaging photoelectron photoion coincidence (i^2^PEPICO) spectrometer at the Swiss Light Source (SLS) [29,30]. A microwave discharge flow tube and a laser photolysis flow tube are employed in Hefei and at the SLS, respectively, to generate radicals and to follow their fate. The dimeric product C_2_H_5_OOC_2_H_5_ formed in the self-reaction of C_2_H_5_O_2_, as well as the major products C_2_H_5_O, CH_3_CHO and C_2_H_5_OH, have been clearly observed and identified in the photoionization mass spectra. Kinetic experiments have also been carried out to confirm the origins of the products and to validate the detailed reaction mechanisms inside the flow tube. In addition, the mass-selected photoionization spectrum or photoionization efficiency curve (PIE) corresponding to C_2_H_5_OOC_2_H_5_ has been measured by scanning synchrotron photon energy, and its adiabatic ionization energy (AIE) is determined at 8.75 eV with the aid of Franck-Condon calculations. In addition, the electronic structure of C_2_H_5_OOC_2_H_5_ is revealed here. Meanwhile, high-level theoretical calculations are also performed and combined to obtain the potential energy surface of the C_2_H_5_O_2_ self-reaction.

## 2. Results and Discussion

### 2.1. VUV Lamp Photoionization Mass Spectra

To obtain the origin of the products and to study the complex chemistry inside the microwave discharge flow tube, the experiments were conducted in sequences. The ethyl radicals (C_2_H_5_) were formed from the reaction of ethane with fluorine atoms, which were generated in the microwave discharge generator with 5% fluorine gas in helium.
F + C_2_H_6_ → C_2_H_5_ + HF(4)

Firstly, the photoionization mass spectrum was measured without adding oxygen into the reactor by using the VUV lamp-based photoionization mass spectrometer [27,28], and is presented in Figure 1a. The most intense peak, at m/z = 29, is assigned to the ethyl radical C_2_H_5_, and other peaks can be ascribed to the products of the self-reaction of the ethyl radicals or secondary reactions in the flow tube [31].

Secondly, abundant oxygen was added into the flow tube and the ethyl radicals reacted with oxygen to generate either the ethyl peroxy radicals (C_2_H_5_O_2_) or ethene (C_2_H_4_) with the hydroperoxyl radicals (HO_2_).
C_2_H_5_ + O_2_ (+ M) → C_2_H_5_O_2_ (+ M)(5)
       → C_2_H_4_ + HO_2_(6)

In the reaction (5), M represents the molecules that remove the internal energy of the nascent C_2_H_5_O_2_ through collisions. Figure 1b presents the photoionization mass spectrum measured with the addition of oxygen. Previous studies [31,32,33,34] have demonstrated that the cation of the ethyl peroxy radical is unstable upon photoionization and dissociates to C_2_H_5_^+^ and O_2_ fragments, so the m/z = 29 mass peak in Figure 1b is contributed from C_2_H_5_O_2_. The peak at m/z = 28 is assigned to the product C_2_H_4_. As the AIE of HO_2_ is located at 11.359 eV [35], above the VUV lamp’s photon energies (hν = 10.0 and 10.6 eV), no signal of HO_2_ radicals (m/z = 33) is observed in the photoionization mass spectrum. The mass peak at m/z = 32 is ascribed to O_2_^+^ from the ionization of oxygen presented with a high concentration in the flow tube.

Under the present experimental conditions, nascent C_2_H_5_O_2_ radicals will mainly perform a self-reaction or react with the HO_2_ radicals in the flow tube. The rate constants of the C_2_H_5_O_2_ self-reaction and the cross reaction with the HO_2_ radicals were measured at (1.0 ± 0.2) × 10^−13^ and (6.2 ± 0.2) × 10^−12^ cm^3^·molecule^−1^·s^−1^, respectively [26,36]. For the self-reaction of C_2_H_5_O_2_, it has been predicted that there are three reaction channels: a termination channel (7), producing the two stable products acetaldehyde (CH_3_CHO) and ethanol (C_2_H_5_OH); a non-termination channel (8), producing two ethoxy radicals (C_2_H_5_O); and the dimeric product C_2_H_5_OOC_2_H_5_ channel (9) [14,17]. The cross reaction of the C_2_H_5_O_2_ radicals with the HO_2_ radicals produces ethyl hydroperoxide (C_2_H_5_OOH) with a yield of 100% [25].
C_2_H_5_O_2_ + C_2_H_5_O_2_ → CH_3_CHO + C_2_H_5_OH + O_2_(7)
       → C_2_H_5_O + C_2_H_5_O + O_2_(8)
      → C_2_H_5_OOC_2_H_5_ + O_2_(9)
C_2_H_5_O_2_ + HO_2_ → C_2_H_5_OOH + O_2_(10)

These products can be clearly observed in the photoionization mass spectrum in Figure 1b, i.e., CH_3_CHO at m/z = 44, C_2_H_5_O at m/z = 45, C_2_H_5_OH at m/z = 46, C_2_H_5_OOH at m/z = 62 and C_2_H_5_OOC_2_H_5_ at m/z = 90. Note that, although some of these products have been detected previously with the techniques of molecular modulation spectroscopy [23,25], Fourier transform infrared spectroscopy (FTIR) [22], cavity ring-down spectroscopy (CRDS) [26] and photoionization mass spectrometry (PIMS) [31], the dimeric product C_2_H_5_OOC_2_H_5_ from the self-reaction of C_2_H_5_O_2_ is directly observed for the first time by using VUV photoionization mass spectrometry.

### 2.2. Kinetics and Reaction Mechanism

To confirm the origin of the above-mentioned products, kinetic experiments have been performed by changing the reaction time inside the microwave discharge flow tube; that is, adjusting the distance between the inner tube of the flow tube and the sampling skimmer of the VUV photoionization mass spectrometer [27,28]. The ion signals of the stable products, CH_3_CHO (m/z = 44), C_2_H_5_OH (m/z = 46), C_2_H_5_OOH (m/z = 62) and C_2_H_5_OOC_2_H_5_ (m/z = 90), were measured at several reaction times and are presented as black solid dots in Figure 2. To reveal the embedded reaction mechanisms inside the flow tube, the time evolution of each product has been calculated with a custom-made program, the Chemical Reactions Simulator V2.6-01 [26,36], and is presented as red solid lines in Figure 2. The major reactions used in the kinetic calculations, as well as their literature rate constants, are listed in Table 1. In the kinetic simulations, the reaction temperature is set at 298 K and the concentrations of the reactants (fluorine atoms, ethane and oxygen) are kept consistent with the experimental conditions. As shown in Figure 2, although there are some differences, the overall trends of the calculated results agree with the experimental data. Note that the experimental concentration of each product in the photoionization mass spectra is less well-defined due to the lack of relevant data, such as the photoionization cross sections, mass discrimination factors and other instrumental factors, which could add to the uncertainty of the measurement.

In Figure 2, we can see that, within the present time range, the ion signals of acetaldehyde (m/z = 44), ethanol (m/z = 46), ethyl hydroperoxide (m/z = 62) and the dimer product C_2_H_5_OOC_2_H_5_ (m/z = 90) increase with the reaction time. Moreover, for the species CH_3_CHO, the kinetic simulations show that, in addition to the major contribution from the C_2_H_5_O_2_ self-reaction (blue dashed line in Figure 2a) in the flow tube, it can also be formed from the reaction of C_2_H_5_O_2_ with the C_2_H_5_O radicals (cyan dashed line) and the reaction of the C_2_H_5_O radicals with O_2_ (green dashed line). The product C_2_H_5_OOH has two sources: the cross reaction of C_2_H_5_O_2_ with HO_2_ radicals (pink dashed line in Figure 2c) and the bimolecular reaction of C_2_H_5_O_2_ with C_2_H_5_O radicals with a minor contribution (cyan dashed line), respectively. As shown in Figure 2d, the product C_2_H_5_OOC_2_H_5_ is almost solely formed from the self-reaction of C_2_H_5_O_2_. Note that the product C_2_H_5_OOC_2_H_5_ might also be formed from the reactions of C_2_H_5_ + C_2_H_5_O_2_ and C_2_H_5_O + C_2_H_5_O in the flow tube. However, the kinetic experiments with different concentrations of oxygen demonstrate that these possibilities should be very small, as shown in Appendix A. Based on the peak area ratios of the photoionization mass spectra and the fitting of the kinetic results, a branching ratio of 10 ± 5% larger than the previous results is suggested, corresponding to the reaction rate constant of the dimeric product C_2_H_5_OOC_2_H_5_ channel of 1.1 × 10^−14^ cm^3^·molecule^−1^·s^−1^, as listed in Table 1 [14,17,21].

To further verify the origin of the dimeric product C_2_H_5_OOC_2_H_5_, another kind of kinetic experiment has been performed in the microwave discharge flow tube by changing the initial concentration of the reactants. In these experiments, the ion counts of C_2_H_5_OOC_2_H_5_ were measured at several different concentrations of fluorine atoms and are presented as black dots in Figure 3, in which the ion counts of C_2_H_5_OOC_2_H_5_ increase with the product of the ion signal of C_2_H_5_O_2_, *I*(C_2_H_5_O_2_) × *I*(C_2_H_5_O_2_), in accordance with the second-order kinetics of the C_2_H_5_O_2_ self-reaction [31]. The concentration of C_2_H_5_OOC_2_H_5_ has also been calculated by using the Chemical Reactions Simulator with the major reactions in Table 1 and is presented as a red solid line in Figure 3. The calculated concentration of C_2_H_5_OOC_2_H_5_ also increases with the product of the concentration of C_2_H_5_O_2_, [C_2_H_5_O_2_] × [C_2_H_5_O_2_], and its overall shape agrees well with the experimental data, demonstrating the reaction mechanism inside the flow tube.

### 2.3. Photoionization Spectrum and Structure

Photoionization experiments were also performed at the Swiss Light Source by using the i^2^PEPICO setup [29,30]. The above-mentioned products, CH_3_CHO, C_2_H_5_O, C_2_H_5_OH, C_2_H_5_OOH and C_2_H_5_OOC_2_H_5_, are also observed in the synchrotron photoionization mass spectra, shown in Appendix A. The photoionization spectrum or the PIE curve (see Figure 4) of the dimeric product C_2_H_5_OOC_2_H_5_ (m/z = 90) has been measured in the energy range of 8.0–10.8 eV with a photon energy step size of 20 meV. The PIE curves of the other species, as well as their mass-selected threshold photoelectron spectra (TPES), can be found in Appendix A. In Figure 4, although the photoionization spectrum is noisy due to the small branching ratio, the appearance of C_2_H_5_OOC_2_H_5_^+^ cations can still be discerned from the background and are located at hν = 8.75 ± 0.05 eV. We also performed a full optimization of the structures of the dimeric product C_2_H_5_OOC_2_H_5_ and its cation C_2_H_5_OOC_2_H_5_^+^ at the explicitly correlated coupled cluster single-double and perturbative triple excitations approach, CCSD(T)-F12, in conjunction with the aug-cc-pVTZ basis set as implemented in the MOLPRO 2015 program [44]. The theoretical calculations show that both the dimeric product C_2_H_5_OOC_2_H_5_ and its cation C_2_H_5_OOC_2_H_5_^+^ have stable structures, as presented in Figure 5. With the exception of the slight shortening of the O-O bond length, from 1.421 to 1.293 Å upon ionization, the configuration of the cation C_2_H_5_OOC_2_H_5_^+^ is similar to that of the neutral C_2_H_5_OOC_2_H_5_. The AIE of C_2_H_5_OOC_2_H_5_ has also been theoretically calculated at the same level of theory and is predicted at 8.795 eV.

In order to assign the photoionization spectrum, the Franck-Condon factors for the ionization transition of the dimeric product C_2_H_5_OOC_2_H_5_ have been calculated at the M06-2X/aug-cc-pVTZ level of theory by using the Gaussian 16 program package [45]. The calculated photoelectron spectrum (PES) is obtained from the Franck-Condon factors by convolving the stick spectrum with a Gaussian function (with its HWHM = 200 cm^−1^, half width at half maximum), as shown in Appendix A. Subsequently, the theoretical photoionization spectrum can be obtained through the integration of the PES, and is presented as a red solid line in Figure 4, with a small energy shift to fit the experimental spectrum. The comparison between the experimental and theoretical photoionization spectra of C_2_H_5_OOC_2_H_5_ at hν = 8.75 ± 0.05 eV and the high-level AIE calculations at 8.795 eV confirms our assignment.

### 2.4. Potential Energy Surface of Self-Reaction

High-level theoretical calculations have also been carried out to understand the detailed processes of the self-reaction of C_2_H_5_O_2_. As illustrated in Figure 6, the potential energy surface of the self-reaction of C_2_H_5_O_2_ has been theoretically calculated and presented, as well as the optimized structures of the reactants, intermediates, transition states (TS) and products. Concretely, the structural optimization of each species is conducted through the density functional theory (DFT) at the M06-2X/aug-cc-pVTZ level using the Gaussian 16 program package [45]. The vibrational frequency calculations show that all of the stable minima have positive vibrational frequencies, and the transition states possess only one imaginary frequency, along the reaction coordinate. Their relative energies with the zero-point vibrational energy corrections are also computed at the same level of theory, with respect to that of C_2_H_5_O_2_ + C_2_H_5_O_2_, and are presented in Figure 6. All the computed structures (xyz coordinates) and energies are displayed in Appendix A.

As shown in Figure 6, the interaction between the two doublet C_2_H_5_O_2_ radicals can interact in both a triplet state and a singlet state. The previous theoretical calculations indicate that the self-reaction of C_2_H_5_O_2_ should go through the singlet RO_4_R′ tetroxide structure, which undergoes two separate bond-cleavage reactions to form a singlet intermediate complex ^1^(C_2_H_5_O···^3^O_2_···OC_2_H_5_), where the O_2_ is in its triplet ground state and the two C_2_H_5_O radicals have parallel spins [16,17]. Thus, the triplet tetroxide structure is not considered here. As the ^3^O_2_ in the complex ^1^(C_2_H_5_O···^3^O_2_···OC_2_H_5_) is only weakly bound, with a distance of ~1.4 angstrom, it will likely evaporate from the system to give a triplet ^3^(C_2_H_5_O···OC_2_H_5_) cluster. The theoretical calculations also predict that the proceeding reactions of the ^3^(C_2_H_5_O···OC_2_H_5_) cluster have three channels: (1) evaporation/dissociation to two C_2_H_5_O radicals; (2) H-shift to give alcohol (^1^C_2_H_5_OH) and carbonyl (^3^CH_3_CHO) products; (3) intersystem crossing (ISC), leading to the formation of the dimeric product C_2_H_5_OOC_2_H_5_. In addition, the reaction channel (2) of the ^3^(C_2_H_5_O···OC_2_H_5_) clusters through the H-shift might give two different sets of products: a triplet carbonyl with a singlet alcohol and a singlet carbonyl with a triplet alcohol. According to our calculations, the formation of the triplet ^3^CH_3_CHO and the singlet ^1^C_2_H_5_OH products are the lowest in energy, which is more favorable compared to the singlet carbonyl and the triplet alcohol. The corresponding transition state (3.97 kcal/mol) of the H-shift reactions of ^3^(C_2_H_5_O···OC_2_H_5_) cluster has been calculated as triplet. As the reaction channel (3) of ^3^(C_2_H_5_O···OC_2_H_5_) to form the dimeric product ^1^C_2_H_5_OOC_2_H_5_ is prevented by the Pauli principle, Valiev et al. and Hasan et al. suggested that the triplet cluster should perform an intersystem crossing (ISC, “spin-flip”) to a singlet surface, then allowing for recombination to form the dimeric product [16,17]. Hasan et al. also calculated the ISC rate constant for the ^3^(C_2_H_5_O···OC_2_H_5_) cluster at 3.5 × 10^9^ s^−1^ [17]. As shown in Figure 6, the formation of the dimeric product C_2_H_5_OOC_2_H_5_ is predicted to be highly exothermic, with an energy of −34.05 kcal mol^−1^, which is energetically favorable and agrees well with the experimental results.

## 3. Materials and Methods

The main configurations of the VUV lamp-based photoionization mass spectrometer in Hefei and the microwave discharge flow tube reactor have been introduced in detail in our recent publications and only a brief description is presented here [27,28,31]. Briefly, the microwave discharge flow tube reactor is used to generate radicals and to study the self-reaction of C_2_H_5_O_2_ under NOx-free conditions. The flow tube mainly consists of a main tube (450 mm length, 16 mm inner diameter) and a movable coaxial inner tube (600 mm length, 4 mm inner diameter). The inner surface of the main tube and the outer surface of the inner tube are coated with halocarbon wax to minimize the wall loss of the radicals. F_2_ diluted to 5% in helium is discharged with a 2.45 GHz microwave discharge generator (GMS-200W, Sariem, France) to produce fluorine atoms and then to initiate the reactions. The pressure inside the flow tube is measured by a diaphragm vacuum gauge and precisely controlled by a closed-loop feedback throttle valve. In the present experiments, the pressure of the flow tube is fixed at 266 Pa. The total gas flux into the flow tube is 500 cm^3^ min^−1^ and the reactants’ initial concentrations are 3.7 × 10^13^ molecules·cm^−3^ for F atoms, 3.1 × 10^14^ molecules·cm^−3^ for C_2_H_6_ and 2.6 × 10^15^ molecules·cm^−3^ for O_2_, respectively.

A home-made VUV photoionization mass spectrometer is employed to online probe the intermediates and stabilized products inside the flow tube [28]. The photoionization mass spectrometer is composed of three vacuum chambers: a source chamber, an ionization chamber and a TOF chamber. The flow tube is connected with the source chamber directly, and a 1 mm diameter skimmer is adopted to sample the species inside the flow tube. Then, the species are photo-ionized with a krypton discharge lamp (PKS 106, Heraeus, Germany, hν = 10.0 and 10.6 eV) and the ions are analyzed with an orthogonal acceleration reflectron time-of-flight mass analyzer. The mass resolution of the VUV photoionization mass spectrometer is M/∆M = 2100 (FWHM, full width at half maximum). Very recently, the VUV photoionization mass spectrometer has been upgraded in some content to achieve a better detection limit, LOD < 0.001 µg/L [28]. In the kinetic experiments, the reaction time is varied by changing the distance between the inner tube and the skimmer.

The synchrotron photoionization experiments are performed at the VUV beamline at the Swiss Light Source. The detailed configurations of the synchrotron beamline and the i^2^PEPICO setup can be found in the literature [29,30]. A 10 Hz pulsed Nd-YAG (213 nm, 16 mJ cm^−2^) laser is used to generate chlorine atoms through the photolysis of oxalylchloride (COCl)_2_, which initiates the radical reactions in a side-sampled flow reactor. The reactor is a 57.4 cm long quartz tube with a 1.27 cm outer diameter, 1.05 cm inner diameter, and a 300 µm pinhole at the halfway-point along the tube. Ethane (0.03 sccm) and oxygen (75 sccm), as well as Ar carrier gas (15 sccm), are also added into the flow reactor (6 mbar) to produce ethyl peroxy radicals and the dimeric products C_2_H_5_OOC_2_H_5_. Photoelectrons and photoions are velocity map imaged onto two position-sensitive delay-line detectors, respectively, and detected in delayed coincidences [46]. In the present experiments, due to the weak signal, only the photoionization spectrum of C_2_H_5_OOC_2_H_5_ is acquired and presented here.

High-level theoretical calculations, consisting of the determinations of the structures of the neutral and ionic species of C_2_H_5_OOC_2_H_5_, the corresponding AIE, the Franck-Condon factors involved in the ionization and the reaction potential energy surfaces, have been performed. Briefly, the structures of the dimeric product C_2_H_5_OOC_2_H_5_ and its cation C_2_H_5_OOC_2_H_5_^+^ have been fully optimized at the explicitly correlated coupled cluster single-double and perturbative triple excitations approach, CCSD(T)-F12, in conjunction with the aug-cc-pVTZ basis set as implemented in the MOLPRO 2015 program [44]. Subsequently, the AIE of C_2_H_5_OOC_2_H_5_ has been theoretically calculated at the same level of theory. The Franck-Condon factors for the ionization transitions are calculated at the M062X/aug-cc-pVTZ level of theory using the time-independent adiabatic Hessian Franck-Condon model in the Gaussian 16 package. The reaction potential energy surfaces (PES) for the C_2_H_5_O_2_ self-reaction, as well as the optimization of the structures of the reactants, intermediates, transition state and products are also calculated at the M06-2X/aug-cc-pVTZ level of theory with the Gaussian 16 package [45].

## 4. Conclusions

In summary, the self-reaction of C_2_H_5_O_2_ radicals has been investigated by using advanced VUV photoionization mass spectrometry in combination with high-level theoretical calculations. Both the major products, CH_3_CHO, C_2_H_5_O, C_2_H_5_OH, and the minor dimeric product, C_2_H_5_OOC_2_H_5_, formed from the self-reaction have been clearly observed in the photoionization mass spectra. To verify the origins of these products and to validate the detailed reaction mechanism inside the fast flow tube reactor, two kinds of kinetic experiments were performed, by either changing the reaction time or the initial concentration of C_2_H_5_O_2_ radicals, and both the experimental data agree well with theoretical results calculated with the Chemical Reactions Simulator. Then, based on the fitting of the kinetic data with the theoretically calculated results and the peak area ratios in the photoionization mass spectra, a branching ratio of 10 ± 5% for the pathway leading to the dimeric product C_2_H_5_OOC_2_H_5_ is suggested. In addition, the adiabatic ionization energy of C_2_H_5_OOC_2_H_5_ is determined at 8.75 ± 0.05 eV in the photoionization spectrum with the aid of the Franck-Condon calculations, and its elusive structure is revealed. The potential energy surface of the C_2_H_5_O_2_ self-reaction has also been calculated to understand the reaction process in detail. This study performs the direct measurement of the dimeric product ROOR in the self-reaction of small RO_2_ radicals. Moreover, as demonstrated here, the advanced VUV photoionization mass spectrometry offers an effective means to probe elusive peroxides with the advantages of soft ionization, sensitivity and universality, and will provide new insights to analyze complex atmospheric reactions in more detail.

## Figures and Tables

**Figure 1 ijms-24-03731-f001:**
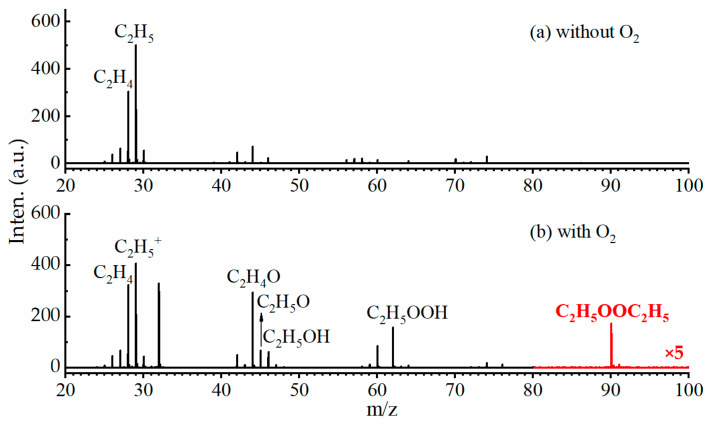
VUV lamp photoionization mass spectra measured (**a**) without O_2_ and (**b**) with O_2_ in the microwave discharge flow tube, with 5 times magnified data in red. The reaction time was 9 ms, and the initial concentrations of F atoms, C_2_H_6_ and O_2_ in the flow tube were 3.7 × 10^13^, 3.1 × 10^14^, and 2.6 × 10^15^ molecules·cm^−3^, respectively.

**Figure 2 ijms-24-03731-f002:**
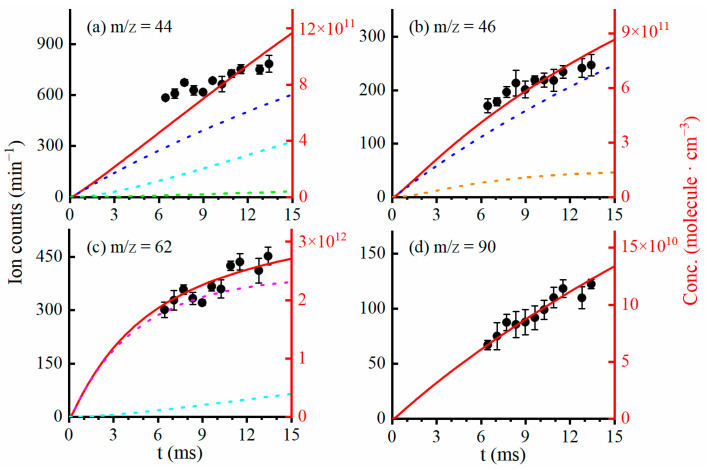
Time-evolutions of products in the self-reaction of C_2_H_5_O_2_ and its cross-reaction with HO_2_. (**a**) CH_3_CHO, m/z = 44, (**b**) C_2_H_5_OH, m/z = 46, (**c**) C_2_H_5_OOH, m/z = 62, (**d**) C_2_H_5_OOC_2_H_5_, m/z = 90. Black dots are experimental data and red curves are calculated from a theoretical model. Blue line: from C_2_H_5_O_2_ self-reaction contribution; cyan line: from C_2_H_5_O + C_2_H_5_O_2_ contribution; pink line: from C_2_H_5_O_2_ + HO_2_ contribution; green line: from C_2_H_5_O + O_2_ contribution; orange line: from C_2_H_5_O + HO_2_ contribution.

**Figure 3 ijms-24-03731-f003:**
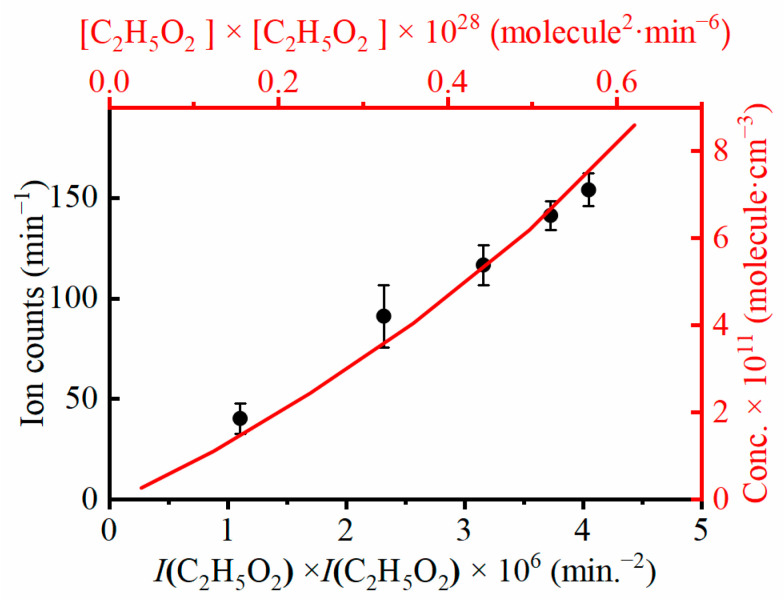
Kinetic of C_2_H_5_OOC_2_H_5_ in the self-reaction of C_2_H_5_O_2_. Black dots are experimental data, and red solid line is calculated results with the reaction mechanism in Table 1.

**Figure 4 ijms-24-03731-f004:**
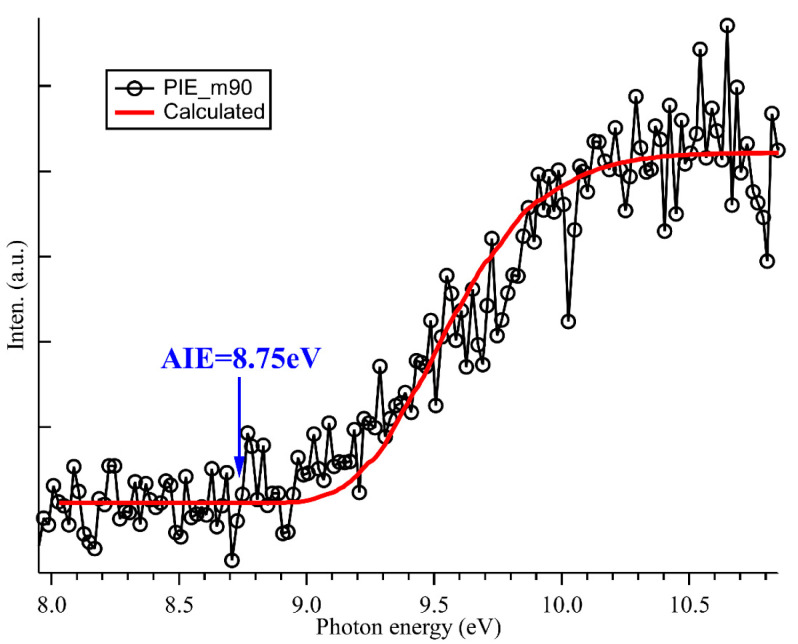
Photoionization spectrum of C_2_H_5_OOC_2_H_5_ and its calculated results in red.

**Figure 5 ijms-24-03731-f005:**
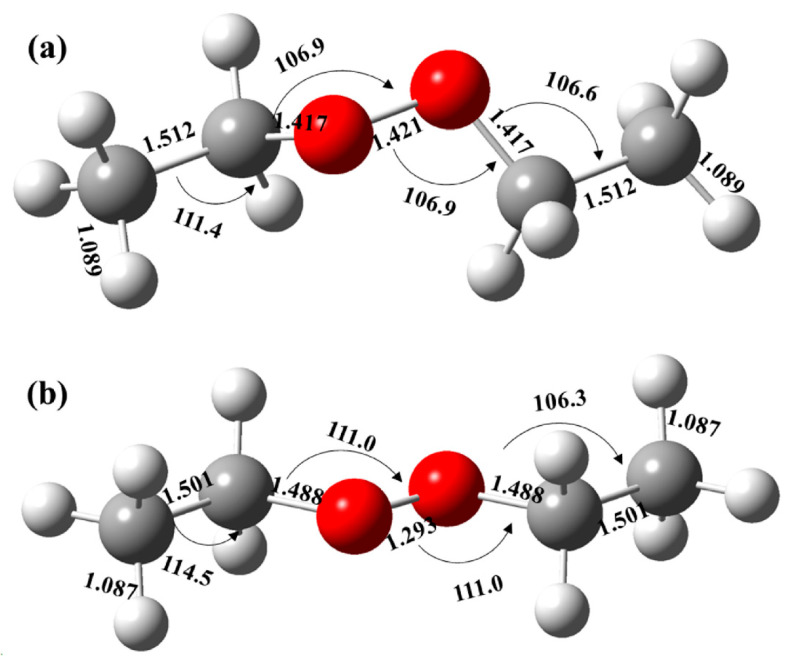
The optimized structures of (**a**) C_2_H_5_OOC_2_H_5_ and (**b**) its cation C_2_H_5_OOC_2_H_5_^+^. The bond lengths in angstrom and bond angles in degrees. Atom colors: O = red; C = gray; H = white.

**Figure 6 ijms-24-03731-f006:**
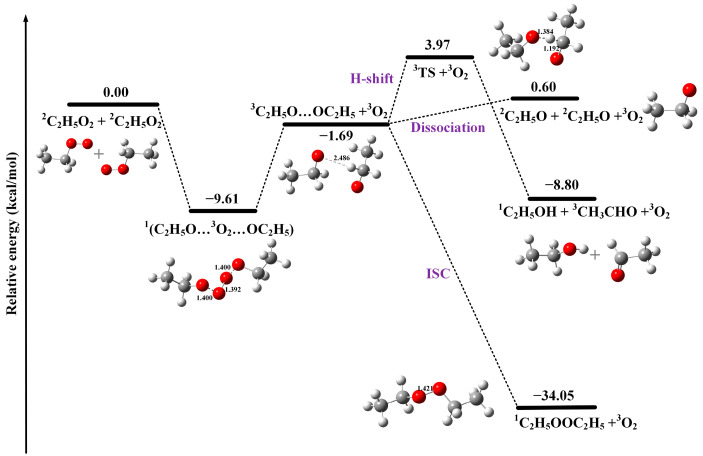
Potential energy surface of the self-reaction of C_2_H_5_O_2_ calculated at the M06-2X/aug-cc-pVTZ level of theory.

**Table 1 ijms-24-03731-t001:** Major reactions used in chemical kinetic calculations.

Lists	Reactions	Rate Constants ^a^	Ref.
1	C_2_H_6_ + F → C_2_H_5_ + HF	1.07 × 10^−10^	[37]
2	C_2_H_5_ + O_2_ + M → C_2_H_5_O_2_ + M	4.80 × 10^−12^	[38]
3	C_2_H_5_ +O_2_ → C_2_H_4_ + HO_2_	4.00 × 10^−13^	[26]
4	C_2_H_5_ + C_2_H_5_ → C_4_H_10_	1.42 × 10^−11^	[39]
5	C_2_H_5_ + C_2_H_5_ → C_2_H_4_ + C_2_H_6_	3.90 × 10^−12^	[40]
6	C_2_H_5_O_2_ + C_2_H_5_O_2_ → 2C_2_H_5_O + O_2_	3.10 × 10^−14^	[26]
7	C_2_H_5_O_2_ + C_2_H_5_O_2_ → C_2_H_5_OH + CH_3_CHO + O_2_	6.0 × 10^−14^	[26]
8	C_2_H_5_O_2_ + C_2_H_5_O_2_ → C_2_H_5_OOC_2_H_5_ + O_2_	1.1 × 10^−14^	This work
9	C_2_H_5_O_2_ + HO_2_ → C_2_H_5_OOH + O_2_	6.20 × 10^−12^	[36]
10	C_2_H_5_O + C_2_H_5_O_2_ → CH_3_CHO + C_2_H_5_OOH	7 × 10^−12^	[26]
11	C_2_H_5_O + O_2_ → CH_3_CHO + HO_2_	8.00 × 10^−15^	[41]
12	C_2_H_5_O + HO_2_ → C_2_H_5_OH + O_2_	1.1 × 10^−10^	[42]
13	C_2_H_5_O_2_ → diffusion	5 ^b^	This work
14	HO_2_ → diffusion	10 ^b^	This work
15	HO_2_ + HO_2_ → H_2_O_2_ + O_2_	1.7 × 10^−12^	[43]

^a^ Unit: cm^3^·molecule^−1^·s^−1^. ^b^ Unit: s^−1^.

## Data Availability

The data presented in this study are available on reasonable requisition.

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
