# Peer review of "Dimeric Product of Peroxy Radical Self-Reaction Probed with VUV Photoionization Mass Spectrometry and Theoretical Calculations: The Case of C_2_H_5_OOC_2_H_5"

_ijms, 2023, doi:10.3390/ijms24043731_

Round 1

Reviewer 1 Report

Tang and co-workers report a mechanistic study of the ethyl peroxy radicals. This reaction is relevant to atmosphere chemistry and its mechanistic channels are relevant to predict the degradation products and its potential damage to the atmosphere. The authors have used several experimental techniques to generate the radicals as isolated as possible and to analyze the kinetics of the reaction. This is later correlated with computational calculations. As a computational chemist, my review is focused in the latest part of the manuscript. The proposed mechanism is reasonable but I miss several key points along the reaction profile that should be added before publication. Overall, I think the manuscript may be suitable for IJMS but, so I would recommend publication after the following major questions/comments:

1-      The multiplicity of the calculations is not clear and should be added to all the points in Figure 6. Also, all the computed structures (xyz coordinates) and energies should be published in the supporting information to ensure the reproducibility of the results.

2-      The reaction starts with two doublets (two C2H5O2 radicals) that can interact in both the triplet state or the singlet state (antiferromagnetic coupling) to release O2. Only one structure has been proposed but the connectivity between the two minima through a transition state in both the triplet and singlet state is missing. This should be calculated and added.

3-      The second step of the reaction (after O2 release) has not been properly simulated. The initial structure is assumed to be triplet but two of the three products (C2H5OH + CH3CHO and C2H5OOC2H5), formed from the corresponding reaction channels are singlet molecules. Therefore, a minimum energy crossing point should take place either before or after the transition states and must be calculated. It is not clear if the calculated transition state (3.97 kcal/mol) has been calculated as singlet or triplet, but the MECP should be added to the mechanism anyway.

4-      Are the relative energies free energies or potential energies? I would add both magnitudes to the profile.

Author Response

Please see our answers to the questions/comments in the uploaded file. Thank you.

Reviewer 2 Report

attached

Author Response

(The authors gave the same response as above.)

Round 2

Reviewer 1 Report

The authors have responded to my previous comments. Despite the ISC of the second step is not calculate, I think the manuscript has been sufficiently improved for publication in IJMS. Also, the multiplicities have been added and the discussion about the first step based on previous literature is more convincing than in the previous version. So, I recommend publication without further changes.